# THEORETICAL AND EMPIRICAL STUDY OF ADVERSARIAL EXAMPLES

## ABSTRACT

Many techniques are developed to defend against adversarial examples at scale. So far, the most successful defenses generate adversarial examples during each training step and add them to the training data. Yet, this brings significant computational overhead. In this paper, we investigate defenses against adversarial attacks. First, we propose **feature smoothing**, a simple data augmentation method with little computational overhead. Essentially, **feature smoothing** trains a neural network on virtual training data as an interpolation of features from a pair of samples, with the new label remaining the same as the dominant data point. The intuition behind **feature smoothing** is to generate virtual data points as close as adversarial examples, and to avoid the computational burden of generating data during training. Our experiments on MNIST and CIFAR10 datasets explore different combinations of known regularization and data augmentation methods and show that **feature smoothing** with logit squeezing performs best for both adversarial and clean accuracy. Second, we propose an unified framework to understand the connections and differences among different efficient methods by analyzing the biases and variances of decision boundary. We show that under some symmetrical assumptions, label smoothing, logit squeezing, weight decay, mix up and feature smoothing all produce an unbiased estimation of the decision boundary with smaller estimated variance. All of those methods except weight decay are also stable when the assumptions no longer hold.

## 1 INTRODUCTION

Machine learning models are often vulnerable to adversarial examples, which are maliciously designed to cause misclassification. In the area of computer vision, for instance, object recognition classifiers are much more likely to incorrectly classify images that have been modified with small, often inperceptible perturbations. Similar problems also occur in natural language processing area, see (Miyato et al., 2017), where small perturbations of text can easily fool a label classification model. It is therefore important to develop machine learning models that are resistant to adversarial examples in situations where attacker may attemp to interfere, for example with autonomous vehicles (Papernot et al., 2017). Understanding the design mechanisms of adversarial examples can also help researchers to gain a better understanding of the performance of machine learning, especially deep learning models. In this paper, we introduce an efficient **feature smoothing** method to improve the adversarial robustness of neural networks and also build a theoretical framework to understand how different approaches help with the adversarial accuracy.

Different adversarial training methods have been proposed to increase robustness by augmenting training data with adversarial examples. Goodfellow et al. (2015) developed the fast gradient signed method (FGSM), which efficiently generated adversarial example by a "single-step" attack based on a linearization of the model's loss. Their trained model is robust to single-step perturbations but remains vulnerable to more costly "multi-step" attacks. Madry et al. (2017) extended FGSM by proposing a multi-step variant FGSM, which is essentially projected gradient descent(PGD). They suggested that adversarial training with the PGD attack is a universal first order adversary defense, which means that models trained against PGD attacks are also resistant against many other first order attacks. Their PGD attacks consists of initializing the search for an adversarial examples at a random point within the allowed norm ball, then running several iterations of the basic iterative method to find an adversarial examples. Kannan et al. (2018) then introduced a logit pairing method (ALP)

which encourages the logits for pairs of examples and their corresponding adversarial examples to be similar. Logit pairing improves accuracy on adversarial examples over trainings based on PGD.

The above successful approaches performed data augmentation by generating adversarial examples during each training step, which will unfortunately bring significant computational burden to the training process. In contrast, more "efficient" training methods without hindering the training speed have also been shown to improve adversarial robustness (In this paper we refer "efficient" methods as data augmentation and regularization methods including mixup, label smoothing, logit squeezing, weight decay, and our proposed feature smoothing). Szegedy et al. (2016) proposed label smoothing, which trains a classifier using soft targets for the cross-entropy loss rather than the hard targets. The correct class is given a target probability of $1 - \alpha$ and the remaining $\alpha$ probability mass is divided uniformly among incorrect classes. Label smoothing reduces overfitting by preventing a network from assigning full probability to each training data, and also offers a small amount of robustness to adversarial examples (Kannan et al., 2018). Kannan et al. (2018) proposed a logit squeezing method which penalizes the logit of each input example. It is showed that combined with adding Gaussian noise into input examples, logit squeezing gave even better results than ALP in some datasets, for example MNIST and SHNV. Zhang et al. (2018) performed data augmentation by training the model on virtual input points as interpolation of two random examples from the training set and their labels, resulting in increasing both the robustness of adversarial examples and the accuracy in clean test data.

In parallel, many theorems have also been proposed to understand the power and existence of adversarial examples. Transferability is shown to be a common property of adversarial examples. Szegedy et al. (2014) and Papernot et al. (2016) found that adversarial examples generated based on a specific neural network can fool both the same neural network trained with different datasets and different neural networks trained with the same dataset. The existence of adversarial examples is still an open question. Possible reasons have been suggested in recent papers, such as low density (Szegedy et al., 2014; Pei et al., 2017), decision boundary too close to the training data (Tanay & Griffin, 2016). However, there are few papers theoretically explaining the similarities and differences between those methods, especially based on their estimation of decision boundaries. Goodfellow et al. (2015) discussed the differences between weight decay and adversarial training by comparing their loss functions in logistic regression, but didn't show how these two methods affect the estimation and accuracy.

The above discussion leaves us two questions:

- Without adding any computational burden during training, these "efficient" methods mainly benefit from data augmentation and regularization, and as a result, resist against adversarial examples to some extent. As most of them are not specifically designed for resisting against adversarial examples, can we develop an "efficient" approach specifically designed to be robust to adversarial examples?
- What are the connections and differences among these "efficient" methods? Can we build a unified framework to analyze them?

Motivated by these two questions, we investigate defenses against adversarial attacks, and the contribution is two-fold. We first propose **feature smoothing**, a data interpolation method that softens the features of input. We show that feature smoothing obtains better performance than other "efficient" approaches on both MNIST and CIFAR10. We also observe the best performance when combining our feature smoothing method and logit squeezing strategy, among all "efficient" methods. We also propose a unified framework to understand how different "efficient" approaches influence the estimation of decision boundary. In particular, based on both simulations and theoretical analysis of logistic regression, we show that under some symmetrical assumption, label smoothing, logit squeezing, weight decay, mixup, feature smoothing and data extrapolation all give an unbiased estimation of boundary with smaller estimation variance. But regularization with weight decay is more sensitive when the assumption may not hold. We believe it is the reason weight decay can hurt the accuracy in clean test data. Our framework are also partially extended to deep convolutional neural networks.

The paper is organized as follows. Section 2 presents our proposed method and other related "efficient" methods. Section 3 reports the performance of feature smoothing against other "efficient" methods. We conduct theoretical analysis and explore the connections and differences among different methods in Section 4. The last section concludes.

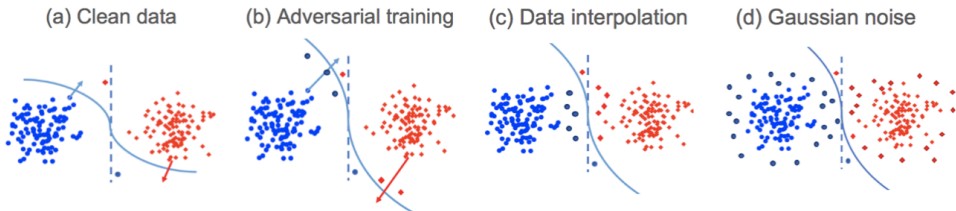

Figure 1: Toy examples of binary classification problem with blue circles and red rhombus representing two classes of data. The blue dash lines and solid lines show true boundary and estimated boundary respectively. (a) The original data; (b) Adversarial examples added into training data; (c) Virtual data interpolated between classes added in training data; (d) Gaussian random noise added into input.

## 2 METHOD

Following the idea of adversarial training, we propose **feature smoothing** method which also adds new data into the training set to improve the robustness. Other than generating adversarial examples based on current model, feature smoothing mimics adversarial examples by data interpolation and adding Gaussian noise directly based on the original training data. We will introduce feature smoothing and discuss several related methods in the following.

### 2.1 FEATURE SMOOTHING

In a classification problem, we aim to recover the unknown decision boundary based on the training data (Figure 1(a)). As long as the decision boundary is correctly estimated, there will be no adversarial examples. Tanay & Griffin (2016) suggested that neural networks which estimate decision boundary too close to the training data causes adversarial problems. The incorrect estimation of boundary may be caused by low density (Szegedy et al., 2014) of input data where adversarial examples exists. In adversarial training, the estimation is improved by adding adversarial examples into input (Figure 1(b)) during each step.

Based on this idea, if we are able to generate 'low density' data directly based on the original training set, we can also improve the estimation as what adversarial training does but with much smaller computational cost. We now introduce **feature smoothing**, a simple data augmentation approach which generates new virtual training data as interpolation of features from a pair of random samples. Virtual training data are constructed as follows:

$$\tilde{\boldsymbol{x}}^{(i)} = (1 - \alpha)\boldsymbol{x}^{(i)} + \alpha\boldsymbol{x}^{(j)}, \qquad \tilde{\boldsymbol{y}}^{(i)} = \boldsymbol{y}^{(i)},$$

where $(\boldsymbol{x}^{(i)}, \boldsymbol{y}^{(i)})$ and $(\boldsymbol{x}^{(j)}, \boldsymbol{y}^{(j)})$ are two examples drawn randomly from our training data, and $0 \leq \alpha < 0.5$. When $\boldsymbol{x}^{(i)}$ and $\boldsymbol{x}^{(j)}$ belong to different classes, and the interval between these two data points intercept with the decision boundary only once, $\tilde{\boldsymbol{x}}_i$ is closer to the boundary than $\boldsymbol{x}_i$ or $\boldsymbol{x}_j$. Figure 1(c) shows that adding new data interpolated between classes can help with the estimation of decision boundary.

Furthermore, Gaussian noise also helps extend the range of $x$. Figure 1(d) shows that adding Gaussian random noise with proper variance into input can also push the estimated boundary closer to the true boundary compared against original clean data. Hence we add Gaussian noise into our feature smoothing method as well:

$$\tilde{\boldsymbol{x}}^{(i)} = P(\boldsymbol{x}^{(i)} + \boldsymbol{\epsilon}), \qquad \tilde{\boldsymbol{y}}^{(i)} = \boldsymbol{y}^{(i)},$$

where $\boldsymbol{\epsilon} \sim Normal(0, \sigma^2)$ and $P(\boldsymbol{x})$ projects $\boldsymbol{x}$ to the range of original data. To distinguish data interpolation part and Gaussian noise part, we use 'feature smoothing' only referring to data interpolation and 'noise' for the Gaussian noise part in the following. A detailed illustration of how feature smoothing helps the estimation of boundary is discussed in Section 4.

## 2.2 RELATED METHODS

Though starting from different intuitions, feature smoothing turns to be very similar with mixup (Zhang et al., 2018). In mixup, additional virtual data points are generated by interpolating both features and labels of the original training data:

$$\tilde{\boldsymbol{x}} = (1 - \alpha)\boldsymbol{x}^{(i)} + \alpha\boldsymbol{x}^{(j)}, \qquad \tilde{\boldsymbol{y}} = (1 - \alpha)\boldsymbol{y}^{(i)} + \alpha\boldsymbol{y}^{(j)},$$

where $\alpha \in [0, 1]$. Mixup can be understood as a form of data augmentation that encourages the model to behave linearly in-between training examples. Zhang et al. (2018) argued that this linear behavior reduces the amount of undesirable oscillations when predicting data outside the training examples. On the contrary, our feature smoothing method includes the interpolations with new label remaining the same as the dominant data point, which maintains the S-shaped curve of logistic model and also allow feature smoothing easier to be combined with regularization methods. More detailed comparison can be found in Sec. 4.

Label smoothing (LaS) and logit squezzing (LoS) are other two efficient ways which improve the adversarial accuracy. Let $y \in \mathbb{R}^K$ be one-hot label for $K$ classes, label smoothing (Szegedy et al., 2016) softens the target by replacing $y$ with

$$\tilde{\boldsymbol{y}} = \frac{\delta}{K - 1}(1 - \boldsymbol{y}) + (1 - \delta)\boldsymbol{y},$$

where $\delta = 0.1$ is shown to be the best choice (Pereyra et al., 2017). Assume we train a model with parameters $\boldsymbol{\theta}$ on a batch of $m$ data points $\{(\boldsymbol{x}^{(i)}, \boldsymbol{y}^{(i)}), i = 1, 2, \ldots, m\}, \boldsymbol{y}^{(i)} \in \{0, 1\}^K$. Let $f(\boldsymbol{x}; \boldsymbol{\theta})$ denote the mapping function from $\boldsymbol{x}$ to logits of the model. Let $L^{(clean)}$ denote the cross entropy loss for the batch of data points as:

$$L^{(clean)} = -\frac{1}{m}\sum_{i=1}^{m}\sum_{j=1}^{K} y_j^{(i)} \log(p_{\boldsymbol{\theta}}(y_j^{(i)}|\boldsymbol{x}^{(i)})).$$

The loss function of label smoothing can also be achieved by some calculation:

$$L_{LaS} = -\frac{1}{m}\sum_{i=1}^{m}\sum_{j=1}^{K} \tilde{y}_j^{(i)} \log(p_{\boldsymbol{\theta}}(y_j^{(i)}|\boldsymbol{x}^{(i)})) = L^{(clean)} - \frac{1}{m}\sum_{i=1}^{m}\sum_{j=1}^{K} \frac{1 - Ky_j^{(i)}}{K - 1}\delta f_j(\boldsymbol{x}^{(i)}, \boldsymbol{\theta}).$$

Notice that if we assume our model obtains a good estimation of $f(\boldsymbol{x}, \theta)$, then when $y_j = 0$, $f_j(\boldsymbol{x}, \theta) < 0$ and when $y = 1$, $f_j(\boldsymbol{x}, \theta) > 0$. In a binary classification case, $L_{LaS}$ can be written as $L^{(clean)} + \delta|f(\boldsymbol{x}, \theta)|$, which further indicates that label smoothing predicts logits with smaller magnitude and therefore avoids overfitting.

Similarly, logit squeezing (Kannan et al., 2018) applies a $L_2$ norm on the logits directly as a penalty of over-confidence:

$$L_{LoS} = L^{clean} + \frac{\lambda}{m}\sum_{i=1}^{m} ||f(\boldsymbol{x}^{(i)})||_2,$$

where $L^{clean}$ is the original loss of neural networks and $f(\boldsymbol{x}^{(i)})$ is the logit of image $\boldsymbol{x}^{(i)}$ as above.

Weight decay is another well known regularizer which efficiently reduces overfitting of neural networks by adding $L_1$ or $L_2$ penalty of weight $w$,

$$L_{wd} = L^{(clean)} + \lambda||\boldsymbol{w}||_2^2.$$

However, weight decay is shown to be not very helpful for adversarial examples compared to label smoothing and logit squeezing, which will be discussed in Sec 4.

**Combination of different approaches**  In feature smoothing and mixup, we generate new data points as linear combination of $x_i$ and $x_j$. For generating the exact $y$ value of these virtual points, mixup uses a linear interpolation for estimation, while feature smoothing chooses the dominant label. Nevertheless, it is also possible that feature smoothing or mixup also adds mislabeled noises into the training data, especially when $\boldsymbol{x}_i$ and $\boldsymbol{x}_j$ are not symmetric to the boundary. In that situation, label smoothing and logit squeezing are better ways to avoid overfitting. So we also consider to combine these methods together to gain a better test and adversarial accuracy.

## 3 EXPERIMENTS

### 3.1 RESULTS ON MNIST

We experiment feature smoothing, label smoothing, mix up, logit squeezing and their possible combinations on MNIST, with the results summarized in Table 1. We find that combining feature smoothing and logit squeezing give the best performance in both clean test data and adversarial examples. For all experiments in this section, we train our models for 200 epochs and use Adam for our optimizer with a learning rate at $10^{-4}$. Random noises are added into the training data in several methods, with the same $\sigma$ value of $0.5$.

For MNIST, when $\alpha$ ranges between $0.2$ and $0.4$ we observe similar performance for feature smoothing, whereas for large $\alpha$ at $0.5$, too much noise in data label brings underfitting for feature smoothing. We use a final $\alpha$ value of $0.3$ for reporting results in Table 1. Chosen by cross validation, we use $\alpha \in Beta(8,8)$ for mixup, and $\delta = 0.1$ for label smoothing. In logit squeezing, we use the weight $\lambda$ of $0.2$ as experimented in Kannan et al. (2018). In feature smoothing and mixup, 10 new data points are generated on each batch with batch size $m = 50$.

We use the LeNet model as Madry et al. (2017) and also apply the same attack parameters as they provided. After scaling the range of images pixels into $[0, 1]$ (divided by 255), we apply perturbation per step of $0.01$, 40 total attack steps with 1 random start, and the total adversarial perturbation threshold set as $0.3$. Similar with Madry et al. (2017), we also generate black box examples for MNIST by independently initializing and training a copy of the LeNet model, then generate PGD attack based on that model. Both cross entropy loss and correct-wrong loss are used.

Each single method improves a small amount of the adversarial accuracy, but combinations of them lead to a much better performance (Table 1). Logit squeezing combined with feature smoothing and Gaussian random noise achieves the best performance among all those "efficient" methods.

| Method | PGD | PGD-cw | BlackBox-PGD | BlackBox-cw | Clean |
|---|---|---|---|---|---|
| Clean | 0.00 | 0.00 | 83.04 | 83.95 | 99.17 |
| Gaussian random noise | 21.02 | 20.34 | 84.57 | 84.26 | 99.20 |
| Logit Squeezing(LoS) | 2.23 | 0.06 | 85.12 | 84.07 | 99.21 |
| Label smoothing(LaS) | 0.06 | 0.08 | 83.13 | 83.07 | 99.22 |
| Mix up | 1.06 | 1.04 | 83.67 | 84.08 | 99.22 |
| Feature smoothing(FS, ours) | 4.68 | 4.76 | 84.54 | 84.98 | 99.26 |
| LoS + noise | 71.49 | 69.04 | 89.01 | 85.34 | 99.22 |
| LaS + noise | 66.26 | 66.62 | 85.99 | 85.26 | 99.24 |
| LoS + noise + mix-up | 70.25 | 68.13 | 84.78 | 83.55 | 99.20 |
| LoS + noise + FS | **76.97** | **75.38** | **90.12** | **89.16** | **99.26** |

Table 1: Accuracies of different methods on MNIST. PGD: projected gradient descent with cross entropy loss; PGD-cw: PGD with correct-wrong loss; Clean: test with original test data. For logit squeezing, we also applied PGD with original cross entropy loss since it gives smaller adversarial accuracy.

### 3.2 RESULTS ON CIFAR10

We follow Madry's lab for the experiments in CIFAR10. For all experiments in this section, we train our models for $80000$ global steps with batch size $m = 128$ in each step. We use Momentum at $0.9$ for our optimizer with a learning rate at $0.1$ for the begining, $0.01$ after $40000$ global steps and $0.004$ after $60000$ global steps. Weight decay with $\lambda = 0.0002$ is also applied to all experiments. We use $\alpha \in Beta(8,8)$ for mixup, $\delta = 0.1$ for label smoothing, $\lambda = 0.1$ for logit squeezing , $\alpha = 0.2$ for feature smoothing, and $\sigma = 0.5$ for Gaussian random noise. In feature smoothing and mixup, 10 new data points are randomly generated on each batch with batch size $m = 128$.

We apply the ResNet model and the same attack parameters as they used. We use perturbation per step of $2.0$, 20 total attack steps with 1 random start and the total adversarial perturbation threshold set as $8$. The black box adversarial examples are also generated by independently initializing and training a same model. Logit squeezing combined with feature smoothing and Gaussian random noise still performs the best among all of the 'efficient' methods (Table 2).

| Method | PGD | PGD-cw | BlackBox-PGD | BlackBox-cw | Clean |
|---|---|---|---|---|---|
| clean | 0.00 | 0.00 | 9.73 | 9.24 | 94.07 |
| LoS + noise | 29.13 | 3.81 | 32.49 | 12.40 | 94.17 |
| LaS + noise | 29.20 | 3.78 | 33.79 | 11.54 | 94.41 |
| LoS + noise + mix-up | 27.49 | 3.85 | 43.22 | 15.24 | 94.62 |
| LaS + noise + mix-up | 27.12 | 3.85 | 43.40 | 15.09 | 94.62 |
| LoS + noise + FS | **32.71** | **9.03** | 48.24 | **17.36** | **95.0** |
| LaS + noise + FS | 30.1 | 5.31 | **49.11** | 16.72 | 94.8 |

Table 2: Top 1 accuracy in CIFAR10.

## 4 THEORETICAL EXPLANATIONS

In this section, we show that the above "efficient" methods increase neural networks' adversarial robustness by improving the estimation of the decision boundaries. The improvement relies on two components: (1) unbiased estimation of boundary; (2) smaller estimation of variance. Given the same training data, these methods estimate the boundary closer to the true boundary than the original neural networks. Our simulations and theoretical results mainly focus on logistic regression. The idea is then discussed with deep convolutional neural networks.

To gain some intuitions on how the above methods improve the estimation, we start from logistic regression model with binary classes. Assume a feature vector $x$ follows some distribution $P_x$ in $\mathbb{R}^d$, $w \in \mathbb{R}^d$ and $b \in \mathbb{R}$, then the corresponding label $y$ follows a Bernoulli distribution with probabilities given by:

$$p := P(y = 1) = \frac{1}{1 + e^{-(wx+b)}}, \qquad P(y = 0) = \frac{1}{1 + e^{wx+b}}. \tag{1}$$

Based on the changing the loss function, we divide these methods into two categories: (1) regularization methods: label smoothing (LaS), logit squeezing (LoS), and weight decay (wd); (2) augmentation methods: mixup and feature smoothing. Regularization methods add penalty term to loss function directly, while augmentation methods modify the loss function by adding new virtual data into it. We analyze the properties of these methods based on the two categories in the following subsections and the proofs of the theorems are included in Appendix.

### 4.1 REGULARIZATION METHODS

Our main theorem shows that all of the regularization methods estimate the decision boundary with smaller variance and the estimation is unbiased when $x$ is symmetric with the boundary. With one-dimensional $x$ and binary classes, the variance of decision boundary can be defined as: $var(\frac{b}{w}), w \neq 0$. Let $\hat{p}$ denote the estimated probability. The confidence interval of $\hat{p}$, which indicates the confidence interval of boundary, is narrowed down with the regularization methods, especially when the support of the distribution of $x$ is far away from boundary (Figure 2(a)). As the value of $w$ increases, the corresponding variances for $w$, $b$ and decision boundary are also better controlled with regularization methods than the vanilla logistic regression (Figure. 3). For the vanilla logistic regression, when $w$ is large enough, the variance of boundary grows in an exponential rate with $w$. But with these regularization methods, variance keeps decreasing even when $w$ is really large. This observation is also true with higher dimensions and multiple classes (Figure 5). Inspired by our observation in the simulation study, we prove the following theorems in one dimension (1-D) to further explain the phenomena in the simulations.

**Theorem 4.1.** *Label smoothing, logit squeezing, and weight decay all estimate the decision boundary with smaller variance in logistic regression model in 1-D.*

**Theorem 4.2.** *When $x$ is symmetric with respect to boundary, label smoothing, logit squeezing, and weight decay have unbiased estimation of boundary in logistic regression model in 1-D.*

The symmetric assumption is not unrealistic for imaging classification problems, since we can always assume the true boundary is in the middle of two classes. However, the stability of those methods when this assumption cannot hold is also important. We also show that label smoothing and logit squeezing is relatively more stable than weight decay when $x$ is asymmetric (Figure 7).

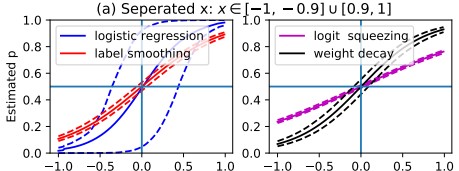 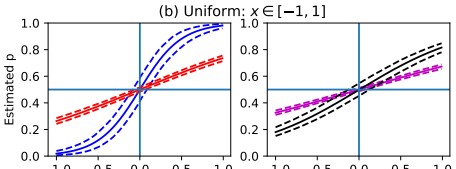

Figure 2: Mean estimated probabilities (solid lines) with $99\%$ confidence intervals (dash lines) obtained from 1000 realizations for regularization methods, with $N = 300$ data points sampled from (a) $Uniform([-1, -0.9] \cup [0.9, 1])$, $w = 4, b = 0$. (b) $Uniform([0, 1])$, $w = 8, b = 4$.

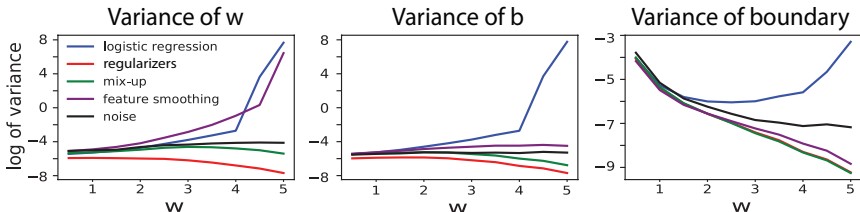

Figure 3: Mean variance of estimated parameters obtained from 1000 realizations with true $w$ increasing from 1 to 6. $x \in Uniform([-1, -0.9] \cup [0.9, 1]$. The legend 'regularizers' represents all regularization methods, since they produce almost identical plots. Here we use $\delta = \lambda = 0.1$ for label smoothing and logit squeezing; $\beta = 0.01$ for weight decay.

## 4.2 AUGMENTATION METHODS

Other than adding regularization to the loss function directly, adversarial training, mixup and feature smoothing all 'improve' the loss function by changing the distribution of $x$. Figure 4 shows how the distribution of $x$ influences the estimation of the boundary in different cases. It is natural to see that when the data are pushed closer to the true boundary, the boundary estimation becomes better due to reduced variances. Following the same analysis above, $x$ being around boundary leads to smaller estimated $p(1 - p)$, which yields smaller variance for $w, b$, and the boundary. When the symmetrical assumption is violated, more careful selection of original data points is needed to avoid adding too much noise into training set.

Following the above explanation, our theorem 4.3 shows that adding data around boundary with labels generated from the true distribution into training can narrow down the variance of boundary, even though the sample size remains the same. Adversarial training, mixup and feature smoothing estimated the labels in different ways. We further show that feature smoothing achieve smaller variance than mixup when $\alpha$ is properly chosen (sec A.4).

**Theorem 4.3.** *Adding data around boundary narrow down the variance of boundary estimation by making the distribution of $x$ closer to boundary. The estimation is unbiased if all labels for the new data are balanced/correctly assigned.*

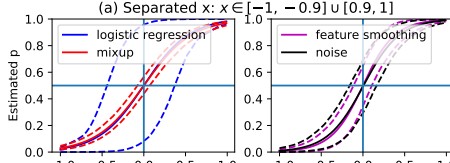 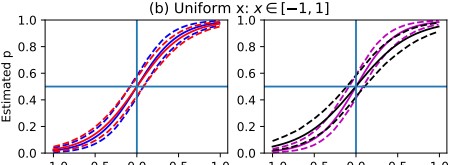

Figure 4: Mean estimated probabilities (solid lines) with $95\%$ confidence intervals (dash lines) obtained from 1000 realizations for augmentation methods, with $N = 300$ data points sampled from (a) $Uniform([-1, -0.9] \cup [0.9, 1])$, $w = 4, b = 0$. (b) $Uniform([0, 1])$, $w = 8, b = 4$. In both cases, we use $\alpha = 0.3$ for feature smoothing and mixup, and $\sigma = 0.3$ for generating random noise.

### 4.3 EXTENSION TO NEURAL NETWORKS

In more complex models like convolution neural network (CNN), the model can be divided into two parts: hidden layers which transform the input data $x \to f(x)$ and the classification model which applies the softmax function (or sigmoid function for binary classification) on $f(x)$. Our results can be extended to CNN for regularization methods since softmax function is just a multi-classes logistic regression. For augmentation methods, we also believe that an interpolation of input data implies an interpolation of transformed data after hidden layers. For simplicity, we assume the nonlinear layers in the CNN only consist of ReLU and max-pooling. Obviously, both ReLU and max-pooling satisfy the following properties: let $\tilde{x} = \alpha x^{(i)} + (1-\alpha)x^{(j)}$, then

$$0 \leq \text{ReLU}(\tilde{x}) \leq \alpha\text{ReLU}(x^{(i)}) + (1-\alpha)\text{ReLU}(x^{(j)}),$$
$$\frac{1}{2}(\alpha\text{max-p}(x^{(i)}) + (1-\alpha)\text{max-p}(x^{(j)})) \leq \text{max-p}(\tilde{x}) \leq \alpha\text{max-p}(x^{(i)}) + (1-\alpha)\text{max-p}(x^{(j)}),$$
$$(2)$$

where max-p represents max-pooling. The first inequality in (2) holds when each argument of $x^{(i)}, x^{(j)}$ is non-negative. Given the pooling layer after the ReLU layer, the assumption is valid. It further implies that

$$f(\tilde{x}) \leq \alpha f(x^{(i)}) + (1-\alpha)f(x^{(j)}),$$

which means augmentation methods on the data can be considered as augmentation on the logits. Then we may use our framework on the logistic regression.

## 5 DISCUSSION

We have proposed feature smoothing, a straightforward data augmentation method as an efficient way to increase adversarial robustness of neural networks. In our experiments, feature smoothing combined with logit squeezing shows the best performance in both MNIST and CIFAR10. We found that $\alpha \in [0.2, 0.4]$ shows similar results when we apply PGD attack with total perturbation threshold as $e = 0.3$. If we use smaller perturbations, smaller $\alpha$, for example $\alpha = 0.1$ for $e = 0.1$, we also observe good results. As a future plan, more possibilities of combinations of different techniques can still be further explored.

We also built a framework to explain how different regularization methods and augmentation methods improve the estimation of decision boundaries for logistic regression. Our main theorems show that all of these methods achieve smaller estimation variance of the decision boundary while keeping the unbiasedness of the estimation. In some extreme cases, for example, correctly labeled data around boundary for one specific class (7), the vanilla logistic regression is incorrectly estimated the boundary for sure but all of the above methods resolve the problem. We also extend the analysis to neural networks based on two facts: (1) the softmax regression is a generalized form of logistic regression in multi-class classification problem; (2)the activation functions like Relu and max-pooling can both keep linear inequalities Eq. (2).

## ACKNOWLEDGEMENT

We would like to thank Dr. Jean-Marc Langlois and Dr. Alyssa Glass for their valuable inputs. We thank Dr. Harini Hannan for providing detailed explanation of her work, so we can successfully replicate experiment results from her paper. We also thank Weiqiang Shi for providing engineering support and Dr. Hua Guo for helpful feedback on drafts of this article.

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

# A  PROOFS

The proofs of Theorem 4.1∼ 4.3 are derived in this section. We first focus on binary logistic regression in the proofs.

## A.1  PROOF THEOREM 4.1

Let $\hat{w}$ and $\hat{b}$ denote the estimates of $w$ and $b$. The decreasing of variance is mainly achieved by two parts: (1) estimated $\hat{w}$ and $\hat{b}$ with smaller magnitude; (2) bias-variance trade-off. We first show that adding regularizers always produce $w$ with smaller magnitude, which lead to smaller variance. Then we show that the bias of $\hat{p}$ introduced by penalties also leads to a smaller variance, essentially when $\hat{p}$ is closer to $0.5$ than the true $p$.

Based on the Fisher's Information, when estimated parameters are MLE, the variances of $\hat{w}$ and $\hat{b}$ are given by:

$$var(\hat{w}) = (E_x x^2 [\hat{p}(1 - \hat{p})])^{-1}, \tag{3}$$

$$var(\hat{b}) = (E_x [\hat{p}(1 - \hat{p})])^{-1}. \tag{4}$$

The decision boundary is $\{x : wx + b = 0\}$. So we use $var(\frac{-\hat{b}}{\hat{w}})$ to measure the variance of estimation of boundary and by delta method

$$var(-\frac{\hat{b}}{\hat{w}}) = \frac{var(\hat{b})}{w^2} + \frac{b^2}{var(1/\hat{w})} + o(1).$$

Without loss of generality, we further assume $b = 0$ and $w > 0$. The variance of $\hat{b}/\hat{w}$ is then equal to

$$\frac{1}{w^2 E_x [\hat{p}(1 - \hat{p})]}.$$

If the distribution of $x$ is a delta mass, i.e., $P_x = \delta_x$, the variance of $\hat{b}/\hat{w}$ can be further written as

$$g(w) = \frac{1}{w^2 \hat{p}(1 - \hat{p})},$$

and the derivative with respect to $w$ is

$$g'(w) = \frac{-2w - w^2 x(1 - 2\hat{p})}{w^4 \hat{p}(1 - \hat{p})}.$$

Given our assumption that $w > 0$, it follows immediately that $x(1 - 2\hat{p}) < 0$ and $wx(1 - 2\hat{p})$ is monotonically decreasing. Moreover, as $w \to \infty$, it yields $wx(1 - 2\hat{p}) \to -\infty$. Therefore, there exists a constant $C$ only depending on $x$ so that for $w > C$ we have $-2w - w^2 x(1 - 2\hat{p}) > 0$. We proved so far that if the estimation is MLE, the variance of boundary is increasing with $w$ when $w > C$.

However, since we add one more regularization term to the original loss function, the estimator is no longer MLE. An approximation of $var(\hat{b})$ is

$$\frac{E_y (y - \hat{p})^2}{\hat{p}^2 (1 - \hat{p})^2},$$

where $y \sim Bernoulli(p_x)$. With regularization methods and based on our assumption of $w$, $0.5 < \hat{p} < p_x$ for $x > 0$ and $p_x < \hat{p} < 0.5$ for $x < 0$. Therefore

$$\frac{E_y (y - \hat{p})^2}{\hat{p}^2 (1 - \hat{p})^2} \leq \frac{1}{\hat{p}(1 - \hat{p})},$$

which indicates a smaller variance of boundary than MLE estimation with the same $\hat{w}$ and $\hat{b}$.

## A.2 PROOF OF THEOREM 4.2

When $\{x\}$ is symmetric to the boundary $x = 0$, the data set can then be splitted into two groups, $\{x_p\}$ containing positive values and $\{x_n\}$ containing negative values, which are symmetric to each other. We further assume the corresponding labels are also approximately symmetric, which is easy to achieve when the sample size is large enough. The loss function is then automatically divided into: $L_p = L(\{x_p\})$ and $L_n = L(\{x_n\})$. The minimizer $\hat{w}$ and $\hat{b}$ of $L_p$ and $L_n$ have to be the same since the input data are symmetric. Then $\hat{w}$ and $\hat{b}$ are also the minimizers of the whole loss function $L = L_p + L_n$. That means for any $x$ in positive part and its corresponding image $x'$ in negative part, the estimated $p_{\hat{\theta}}(y|x) + p_{\hat{\theta}}(y'|x')$ is equal one, which indicates to an unbiased estimation of the boundary.

## A.3 PROOF OF THEOREM 4.3

As mentioned in section 2, new data points

$$\tilde{x} = \alpha x^{(i)} + (1 - \alpha)x^{(j)},$$

are added as input in both methods, while the corresponding $\tilde{y}$ are estimated by either linear interpolation or dominant point of $x^{(i)}$ and $x^{(j)}$. Let $\tilde{p}$ denote the true probability of $\tilde{x}$, we claim that adding $\tilde{x}$ and $\tilde{y} \sim Bernoulli(\tilde{p})$ into input decreases the variance of boundary.

Same as above, the variance of boundary can be estimated as

$$var(-\frac{\hat{b}}{\hat{w}}) = \frac{1}{w^2 \hat{p}(1 - \hat{p})},$$

where $x$ is generated from a distribution $P_x$. When $\tilde{y}$ are generated from the true probability, the MLE estimation do not change. On the other hand, if $x^{(i)}$ and $x^{(j)}$ are from different classes and symmetric to the decision boundary, $\tilde{x}$ is closer to the boundary than $x^{(i)}$ and $x^{(j)}$, and therefore $\tilde{p}(1 - \tilde{p}) > p_{x^{(i)}}(1 - p_{x^{(i)}})$.

## A.4 REMARK OF THEOREM 4.3

As mentioned above, given $x$ from a delta mass distribution, the first derivative of variance with repect to $w$ is given by:

$$g'(w) = \frac{-2w - w^2 x(1 - 2\hat{p})}{w^4 \hat{p}(1 - \hat{p})}.$$

Without loss of generality, we still assume $x > 0, b = 0, w > 0$, then $g'(w) = 0$ gives $w = \frac{2}{x(1-2\hat{p})}$. $g'(w) < 0$ if $w < \frac{2}{x(1-2\hat{p})}$ while $g'(w) > 0$ otherwise. $\tilde{x}$ close to the boundary (0 when $b = 0$) also leads to probability $\tilde{p}$ close to 0.5, i.e. (1 - 2) close to 0. Therefore in a large range of $w \in (0, \frac{2}{x(1-2\hat{p})})$, the variance is decreasing with $w$. As a results, feature smoothing gives even smaller variance than the original MLE with $\tilde{x}$.

## B SIMULATION RESULTS

### B.1 LOGISTIC REGRESSION WITH HIGH-DIMENSIONAL FEATURE AND MULTIPLE CLASSES

### B.2 IN-SYMMETRIC FEATURE

However, in real world, we can never have the perfect scenario that $\{x\}$ is strictly symmetric distributed with respect to the boundary. We further argue that regularizers based on $wx + b$ including label smoothing and logit squeezing are more tolerant to unbalanced data than weight decay regularized on $w$ only. Figure 6 shows how these four methods perform when $\{x\}$ are not symmetrically generated, in two scenarios: (a) data size is unbalanced with respect to decision boundary; (b) data distribution is unbalanced with respect to boundary. It is easy to see that label smoothing and logit squeezing are less sensitive to the distribution of $x$ in both scenarios. In contrast, vanilla logistic regression and weight decay are more sensitive. Confidence intervals for vanilla

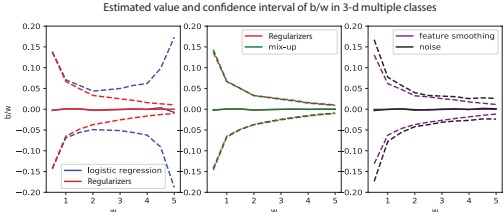

Figure 5: The confidence interval of averaged 'boundary' $\frac{1}{K} \sum_{i=1}^{K} b/\boldsymbol{w}_i$ with $\boldsymbol{x} \in \mathbb{R}^2$ and $K = 3$.

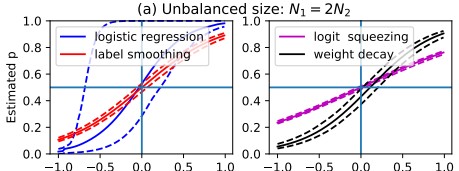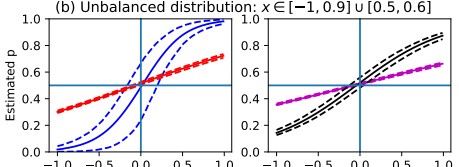

Figure 6: Mean estimated probabilities (solid lines) with $95\%$ confidence intervals (dash lines) obtained from 1000 realizations in the case of in-symmetric data. Data were generated from model 1 with $w = 4, b = 0$. (a) $N_1 = 200$ and $N_2 = 100$ data points were sampled from $[-1, -0.9]$ and $[0.9, 1]$ respectively; (b) both $N_1 = 150$ and $N_2 = 150$ data points were sampled from $[-1, -0.9]$ and $[0.6, 0.7]$ respectively.

logistic regression become wider and do not behave consistently as $x$ value changes; estimated mean decision boundary ($\hat{p} = 0.5$) for weight decay deviate from the true one, not as robust as other methods.

## B.3 ANOTHER REALISTIC CASE

Now let us consider another data unbalance scenario under a different data generation mechanism and see how different methods perform. Note that all of our analysis above assumed that the data label $y$ given data feature $x$ were all generated from the true model 1. In other words, $y$ are random numbers following Bernoulli (Multinomial for multi classes) distribution. Now let us consider another data generation mechanism which is also quite common in real world. Given input $x$, $y$ is deterministic by an identity function $y = I(wx + b)$ instead of following a distribution.

But some classes have data around boundary and some do not, i.e. the distribution of $x$ is unbalanced, for example most of $x \in [-1, -0.9] \cup [0.9, 1]$ but some $x \in [0, 0.1]$. Vanilla logistic regression fails to detect the true boundary in this case but both regularization methods and augmentation methods can improve the estimation (Figure 7).

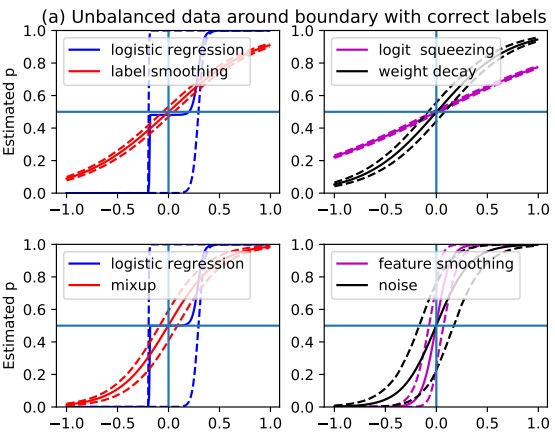

Figure 7: Mean estimated probabilities (solid lines) with $95\%$ confidence intervals (dash lines) obtained from 1000 realizations in the case of in-symmetric data. $x \in [-1, -0.9] \cup [0.9, 1]$ with size 300. $y = 1 \leftarrow p > 0.5$. Then 10 another input $x' = 0.1, y' = 1$ or 10 input of $x' = -0.1, y' = 0$ are added randomly with probability 0.5 to the training.

