# OpenReview forum: "Theoretical and Empirical Study of Adversarial Examples"
_ICLR.cc/2019/Conference_

### Official Review · AnonReviewer1 · 2018-10-31

**Rating:** 4
**Confidence:** 4

**Review:**

The paper proposes a feature smoothing technique, which generates virtual data points by interpolating the input space of two randomly sampled examples. The aim is to generate virtual training data points that are close to adversarial examples. Experimental results on both MNIST and Cifar10 datasets show that the proposed method augmented with other regularization techniques are robust to adversarial attacks and obtain higher accuracy when comparing with some testing baselines. Also, the paper presents some theoretical analyses showing that label smoothing, logit squeezing, weight decay, Mixup and feature smoothing all produce small estimated variance of the decision boundary when regularizing the networks.

The paper is generally well written, and the experiments show promising results. Nevertheless, the proposed method is not very novel, and the method is not comprehensively evaluated with experiments.

Major remarks:

1.	The experiments show that feature smoothing has to combine with other regularizers in order to outperform other testing methods. In this sense the contribution of the feature smoothing along is not clear. For example, without integrating other regularizers, Mixup and feature smoothing obtain very close results for BlackBox-PGD, BlackBoxcw and Clean, as shown in Table 1. In addition, in the paper, the feature smoothing along is only validated on the MNIST (not even tested on Cifar10 in Table2). Consequently, it is difficult to evaluate the contribution of the proposed smoothing technique.
2.	Experiments are conducted on datasets MNIST and Cifar10 with small number of target classes. Empirically, it would be useful to see how it performs on more complex data set such as Cifar100 or ImageNet.
3.	The argument for why the proposed feature smoothing method works is presented in Theorem4.3 in Section 4.2, but the theorem seems to rely on the assumption that one can add data around the true decision boundary. However, how we can generate samples near the true decision boundary and how we should chose the mixing ratio to attain this goal is not clear to me in the paper. In addition, how we can sure that the adding synthetic data from one class does not collide with manifolds of other classes as suggested in AdaMixup (Guo et al., MixUp as Locally Linear Out-Of-Manifold Regularization)? This is particular relevant if the proposed feature smoothing strategy prefers to create virtual samples close to the true decision boundary.
4.	At the end of page4, the authors claim that both feature smoothing and Mixup generate new data points that are closer to the true boundary. I wonder if the authors could further justify or show that either theoretically or experimentally.
5.	The proposed method is similar to SMOTE (Chawla et al., SMOTE: Synthetic Minority Over-sampling Technique). In this sense, comparison with SMOTE would be very beneficial.

Minor remarks:

1.	In the paper Mixup, value 1 was carefully chosen as the mixing policy Alpha for Cifar10 (otherwise, underfitting can easily occur as shown in AdaMixUp), and it seems in the paper the authors used a very large value of 8 for Mixup’s Beta distribution, and I did not see the justification for that number in the paper.
2.	Typo in the second paragraph of page2: SHNV should be SVHN

---

### Official Review · AnonReviewer2 · 2018-11-01
**An interesting paper whose novelty seems incremental to the reviewer**

**Rating:** 5
**Confidence:** 4

**Review:**

The authors proposed a feature smoothing method without adding any computational burden for defensing against adversarial examples. The idea is that both feature smoothing and Gaussian noise can help extend the range of data. Moreover, the authors combined these methods together to gain a better test and adversarial accuracy. They further proved 3 theorems to try to analyze the biases and variances of decision boundary based on the fisher information and delta method.

In my opinion, the main contribution of this paper is to prove that the boundary variance will decrease due to adding one additional regularization term to the loss function.

Main comments:
1.	The proposed feature smoothing method seems less novel to me. In contrast to the mixup method, the proposed method appears to remove the label smoothing part, so it is better to explain or justify why this could be better theoretically.  Moreover, in the PGD and PGD-cw results, the performance is not as good as the Gaussian random noise method. Can the authors offer any discussion or comments on the possible reasons?
2.	Some details of the proof of Theorem 4.1 seemed to be omitted. I am a bit confused about this.
a.	“Without loss of generality, we further assume b = 0 and w > 0.”  With smaller magnitude, b=0 is reasonable, but why to assume w>0?
b.	Could you present the derivation details or the backing theory of the approximation of var(b), when one more regularization term are added?
3.	In addition, a method of modifying the network is proposed to adapt to the feature smoothing method. However, no experimental results are reported to support its effectiveness. I would believe some empirical evaluations may further strengthen the paper.

---

### Official Review · AnonReviewer3 · 2018-11-05
**Some interesting proposals, with weak justification and experimental verification.**

**Rating:** 5
**Confidence:** 2

**Review:**

In this paper the authors introduce a novel method to defend against adversarial attacks that they call feature smoothing. The authors then discuss feature smoothing and related “cheap” data augmentation-based defenses against adversarial attacks in a nice general discussion. Next, the authors present empirical data comparing and contrasting the different methods they introduce as a means of constructing models that are robust to adversarial examples on MNIST and CIFAR10. The authors close by attempting to theoretically motivate their strategy in terms of reducing variance of the decision boundary.

Overall, I found this paper pleasant to read. However, it is unclear to me exactly how novel its contributions are. As discussed by the authors, there are strong similarities between feature smoothing and mixup although I did enjoy the unifying exposition presented in the text. It also seems as though the paper suffers from some simplifying assumptions considered by the authors. For example, in sec. 2 the authors claim that \tilde x will be closer to the decision boundary than x. However, this is only true if the decision boundary is convex.

I appreciated the extensive experiments run by the authors. However, I wish they had included results from adversarial training. It seems (looking at Madry’s paper) that the defense offered by these cheap methods is still significantly worse than adversarial training. I feel that some discussion of this is warranted even if the goal is to reduce computational complexity.

Finally, I am not sure what to make of the theory presented. While it is nice to see that the variance of the decision boundary is reduced by regularization in the case of 1-dimensional linear regression, I am not at all convinced by the authors generalization to neural networks. In particular, their discussion seems to only hold for one-hidden-layer networks. Although the authors don’t offer much clarity here. For example eq. 2 is literally just a statement that ReLU is a convex function. However, it is clearly the case that multiple layers of the network will violate this hypothesis. Overall, I did not find this discussion particularly compelling.

---

### Public Comment · (anonymous) · 2018-09-29
**Interesting but the results seem not good enough**

An interesting method. But have you noticed this work (https://arxiv.org/abs/1705.07204), which proposes a very simple binarization solution on MNIST? Therefore, achieving robustness against l_\infty attacks on MNIST is very simple. And the results on CIFAR10 are not good enough. It seems the accuracy against PGD-cw (LS-PGA) is only 9.03%?
LS-PGA means Logit Space Projected Gradient Ascent (PGD in logit space => PGD using a CW_\infty loss => PGD-CW (l_\infty-bounded))(Just an explanation for the next comment)
For comparison, MadryLab's model achieves 44.71% under DAA and 45.21 under 10 random start PGD. Besides, as you mentioned, this method is more efficient. So have you ever tested on large datasets like ImageNet?

---

> ### Public Comment · (anonymous) · 2018-09-29
> **EAT does not solve MNIST**
>
> I don't disagree with your general sentiment, but two minor notes:
> - Ensemble Adversarial Training itself does not claim to solve MNIST in the white-box setting. It is in Appendix C.1 where the authors note that binarization for L_infinity MNIST is effective.
> - LS-PGA doesn't have anything to do with their attack method (and I don't think they claim it does). PGD-CW is (if I understand correctly, becuase this paper doesn't explain it) PGD from Madry et al. (2018) with the loss function from Carlini & Wagner (2017).
>
> One more observation about the paper: BlackBox-CW on CIFAR-10 accuracy (17%) should not be a stronger attack and have have lower accuracy than than White Box PGD (32%).

---

> > ### Public Comment · (anonymous) · 2018-09-29
> > **Exactly, what I mean is the solution in the appendix.**
> >
> > PGD-CW use the CW_\infty loss, which is actually similar to LS-PGA (PGD using logits, PGD in logit space => PGD using a CW_\infty loss => PGD-CW (l_\infty-bounded))

---

> > > ### Public Comment · (anonymous) · 2018-09-29
> > > **LS-PGA terminology**
> > >
> > > This previous ICLR paper (https://openreview.net/pdf?id=S18Su--CW) proposes an attack called LS-PGA, which is what I thought you were referring to. If you just mean PGD on the logits, then yes, agreed.

---

> > > > ### Public Comment · (anonymous) · 2018-09-29
> > > > **My bad (I did not explain all the things clearly)**
> > > >
> > > > Their LS-PGA seems specific to their thermometer method. When I do experiments on other defenses, I always refer to PGD on CW_\infty as Logit space PGD(PGA).  My personal habit, this is my bad, I should explain it clearly.
> > > > I also agree with your other comments.

---

### Public Comment · ~Marius_Mosbach1 · 2018-11-05
**Cheap methods are unlikely to be robust under stronger PGD attack**

Regarding your results on MNIST, I would like to point you to Table 1 in our paper https://arxiv.org/abs/1810.12042 where we show that logit squeezing (combined with gaussian noise), as proposed by Harini et al., does not provide actual robustness. We could successfully break it not only on MNIST but also CIFAR10 and Tiny ImageNet. Further, we find that the robustness of logit squeezing mainly comes from the fact that it makes gradient based optimization in the input space significantly more difficult by introducing many local maxima near the clean inputs. This can be seen as gradient masking. Crucial for our evaluation was the fact that we performed many random restarts when performing PGD (up to 10000) and additionally performed a proper grid search over the step size used during optimization.

Therefore, it would be interesting to see the robustness of your models against a PGD attack with large number of iterations, large step size, and many random restarts. Based on our experiments, we would expect that this should reduce the adversarial accuracy of "cheap methods" (logit squeezing + noise, label smoothing + noise, feature smoothing + noise)  down to (almost) 0%.

---

> ### Public Comment · (anonymous) · 2018-11-05
> **Sweeping generalizations**
>
> Hello,
> Please refrain from making sweeping generalizations. Is there a proof that "Cheap methods are unlikely to be robust under stronger PGD attack"? The analysis in your paper that you cite is limited to logit squeezing.. why does this imply that "any" cheap method is likely to not be robust?

---

### Meta-Review · Area_Chair1 · 2018-12-15
**Interesting proposal but requires more comprehensive evaluation and comparison**

**Confidence:** 5
**Recommendation:** Reject

**Metareview:**

The paper proposes a feature smoothing technique as a new and "cheaper" technique for training adversarially robust models.

Pros:

* the paper is generally well written and the claimed results seem quite promising

* the theory contribution are interesting

Cons:

* the main technique is fairly incremental

* there were concerns regarding the comprehensiveness of evaluations and baselines used